# Blood Pressure Correlates with Serum Leptin and Body Mass Index in Overweight Male Saudi Students

**DOI:** 10.3390/jpm13050828

**Published:** 2023-05-13

**Authors:** Shalan Alaamri, Abdulhalim S. Serafi, Zahir Hussain, Munira M. Alrooqi, Mohammed A. Bafail, Sumera Sohail

**Affiliations:** 1Department of Medicine, College of Medicine, University of Jeddah, Jeddah 21589, Saudi Arabia; shalaamri@uj.edu.sa; 2Department of Physiology, Faculty of Medicine, Umm Al-Qura University, Makkah 21955, Saudi Arabia; zhakbar@uqu.edu.sa (Z.H.); mabafail@uqu.edu.sa (M.A.B.); 3Department of Chemistry, Faculty of Science, Umm Al-Qura University, Makkah 21955, Saudi Arabia; mmrooqi@uqu.edu.sa; 4Department of Physiology, University of Karachi, Karachi 75270, Pakistan; sumera_sohail@yahoo.com

**Keywords:** systolic blood pressure, diastolic blood pressure, serum leptin, body mass index, normal- and overweight male students, serum apelin

## Abstract

The precise association of serum leptin (Lep) with the body mass index (BMI) and blood pressure (BP) is not well known for understanding their involvement in health and disease. Hence, the present study was conducted to investigate the association of BP, BMI and serum Lep levels in young normal-weight (NW) and overweight (OW) male Saudi students. The NW (n: 198) and OW (n: 192) male subjects in the age range of 18–20 years were consulted. The BP was measured with a mercury sphygmomanometer. Leptin Human ELISA Kits were employed for the determination of the serum Lep levels. The mean ± SD values of BMI (kg/m^2^), Lep (ng/mL), systolic BP (SBP; mmHg), and diastolic BP (DBP; mmHg) all showed significant differences for young OW vs. NW subjects as: 27.52 ± 1.42 vs. 21.49 ± 2.03; 10.70 ± 4.67 vs. 4.68 ± 1.91; 121.37 ± 2.59 vs. 118.51 ± 1.54 and 81.44 ± 1.97 vs. 78.79 ± 1.44, respectively. All associations (among BMI, Lep, SBP and DBP) showed a positive linear and significant correlation, except the nonsignificant correlation of BMI and SBP for the NW group. Other variables showing significant variation for NW vs. OW subjects were: interleukin-6, high sensitivity C-reactive protein, apelin (APLN) and resistin. Serum APLN correlated significantly with Lep, BMI, SBP and DBP in lower and higher levels of BMI, with considerable progressive patterns in both the NW and OW groups and subgroups. The present study in young Saudi male students presents significant variations for BP and serum leptin levels, and a significant positive linear association among serum leptin, BMI and BP.

## 1. Introduction

Leptin is an important adipokine, used as an important clinical marker to investigate various levels of the body mass index (BMI) in normal-weight (NW), overweight (OW), obese and non-obese subjects. Obesity-related hypertension (ORH) and early onset essential hypertension (EH) have been shown to be associated with serum leptin levels [1,2,3]. Leptin (Lep) has also been found to be related to obesity–hypertension syndrome [4], metabolic syndrome (MetS) [5], conditions with moderate association with BMI [6] and several other conditions [7,8,9,10]. The detailed review studies reveal comprehensive information about the role of Lep in cardiovascular disorders and other diseases [6,11,12,13,14].

A positive association of serum Lep and BMI was obtained [15,16,17,18]. It was noted that BMI was significantly and independently associated with increased levels of serum Lep [17,19,20] or obesity-associated blood pressure (BP) changes [1]. Overweight children and adolescents had a significantly greater BMI independently associated with elevated Lep levels [17].

BP was also found associated with Lep levels [8,10,15], specifically systolic blood pressure (SBP) [15] or diastolic blood pressure (DBP) [21]. However, controversial results for the relationship of Lep levels, and BMI and BP were also found [6,10]. Concerning the ethnic background, higher levels of Lep and BP in Africans than Caucasians were revealed [22]. The BP elevation, especially in men, irrespective of black and white race, was associated with serum Lep levels [23].

Subjects having a high–normal and high BP indicated elevated BMI and serum Lep, and the male adolescents showed that serum Lep levels significantly correlating with BMI, SBP and the mean BP [15]. Leptin is an independent mediator for an obesity-related increase in BP, as significantly higher mean BMI, SBP and DBP are associated with Lep levels [19,24,25]. Leptin plays a crucial role in obesity-related hypertension, especially in adolescents where BMI and serum Lep increase according to the order of BP categories (normotensive < high normal < hypertensive), and a significant correlation with both SBP and DBP exists even after making adjustment for the age and BMI [26].

No significant correlation of Lep with SBP or DBP was found [16]. Leptin levels in non-obese normal-weight subjects and obese subjects showed a significant and positive correlation with BMI, but no correlation with the mean SBP or mean DBP [16]. Another study revealed that no statistically significant association of Lep with SBP or DBP was obtained among the subjects having a higher BP, and it was understood that Lep may balance and elevate the arterial tone for increasing DBP only within the normal BP limit [21], and it was concluded that Lep might serve as a physiological mediator or at least a potential marker for increasing the DBP in obesity conditions [21]. 

It was found that Lep associates with DBP, but not with SBP in men with a mean BMI of 23 kg/m^2^ [21]. It was further confirmed that when the subjects were classified as those with a normal BP (SBP < 130 mmHg and DBP < 85 mmHg) and those with a higher BP, Lep showed a significant positive association with DBP, but not with SBP for subjects with a normal BP range [21]. Another contradictory report reveals an association of Lep with BMI instead of BP [27], where it was explained that BMI and BP depend on various other parameters as well, including age, sex, race and the type of obesity [11].

In view of the mentioned reports, it is still controversial to suggest the precise relationship between serum Lep, BP (SBP or DBP) and BMI for the Saudi population. Hence, it is necessary to carry out further studies for clarifying the role of BMI and BP in association with serum Lep in normal-weight and overweight young subjects.

Furthermore, we found a number of studies describing the association of leptin with obesity, but a small number of studies investigating the role of leptin in normal healthy, and moderate- and high-overweight people. Hence, we carried out the present study to further understand the involvement of leptin and its association with the body mass index, blood pressure and other characteristic factors.

## 2. Materials and Methods

### 2.1. Subjects and Study Design

The present study comprised young normal-weight (NW) male university students (n: 198; age range: 18–20 years) and overweight (OW) male university students (n: 192; age range: 18–20 years). General/clinical, anthropometric, physiological, biochemical and cardiovascular measurements were obtained, and the records filled in a comprehensive questionnaire. The research work was carried out at the Umm Al-Qura University, Makkah, Saudi Arabia, from June 2021 to December 2022. The sample size was calculated using the sample size calculator. The subjects were characterized, and the data were recorded and analyzed statistically for body mass index (BMI, kg/m^2^), blood pressure (BP, mmHg; SBP and DBP), serum Lep levels (ng/mL) and related variables. 

The present research work is an observational study for the quantitative observations/correlation analysis of serum Lep levels with SBP and DBP, based on BMI and BP. The subjects selected for the present part of the study were young NW and OW male university students having no serious medical disorders. Only unmarried, non-smoking subjects were accepted for this study. The questionnaire for each subject was filled with detailed physical, psychological and socio-cultural information. Age, educational level, years of education, ethnicity, temperament, personal habits, nutrition and other relevant information were obtained for initially assessing the general health of the subjects and decide whether to include or exclude for the current part of our research. Some of the initial studies served as pilot studies.

The subjects were well informed during their first visit that the subjective information, blood analysis and other tests/examination in the study are for research purpose. It was conveyed to them that it was their decision whether they were willing to take part in the present study. Detailed information through an interview/assessments of each subject was taken after obtaining the subject’s consent for participating in the study. 

### 2.2. Assessments and Methods

Total cholesterol (TC) was measured employing the enzymatic colorimetric method and routine kit methods. A general method was used for determining the fasting blood glucose (FBG) [28]. A specific method for the determination of serum triglycerides (TG) [29] was used. Body weight (kgs)/body height (m^2^) were assessed for the estimation of the body mass index (BMI). The body mass index (BMI) of a person (expressed in kg/m^2^) is defined as the body mass (weight in kilograms) divided by the square of body height (meters) [30,31]. The normal-weight and overweight subjects for the present study had the BMI range of 18 to <25 (kg/m^2^) and >25 to 30, respectively, for the Saudi population [32]. The subjects having higher or lower values for BMI than mentioned were not included. 

The blood pressure levels of the subjects were determined to know whether these levels were normal [33] or exceeding the normal [34]. A general routine method using a BP apparatus was employed for measuring systolic and diastolic blood pressure [35]. The BP was measured with a mercury sphygmomanometer (MS-S1500 Mercury Sphygmomanometer, Medical Sources Co., Limited, Nanjing, Jiangsu, China). 

Blood samples were taken, and the serum Lep was determined. ELISA kits were used for the determination of serum Lep, interleukin-6 (IL-6), high-sensitivity C-Reactive protein (hsCRP), homocysteine (Hcy), hepcidin (Hp), adiponectin (APN), apelin (APLN) and resistin (RETN). Leptin Human ELISA Kits were purchased for the determination of the serum Lep levels. The mean ± SD for BMI (kg/m^2^), serum Lep (ng/mL), SBP (mmHg) and DBP (mmHg) and other variables were evaluated. 

### 2.3. Statistical Analysis

Statistical Package for Social Sciences (version 24.0) for Windows (SPSS Inc., Chicago, IL, USA) was used for statistical analysis. The GraphPad Prism (version 6.0) software (San Diego, CA, USA) was also employed. Data were expressed as the mean ± standard deviation (SD). The distribution of quantitative characteristics corresponded to the normal one. A one-way analysis of variance (ANOVA) with Tukey’s Kramer post hoc test, and the Student’s t-test were carried out. Regression lines were plotted and the values of R^2^ (coefficient of determination) for the regression lines were assessed for obtaining the correlation. The value of significance (*p*) for a linear relationship was obtained. The minimum level of significance was considered *p* ≤ 0.05. Previously published biostatistical applications were followed for statistical analysis [36]. 

## 3. Results

The general characteristics and variables of OW and NW study subjects are shown in Table 1. No significant difference (*p* > 0.05) was obtained for age (years), FBG (mg/dL), TC (mg/dL), TG (mg/dL), Hcy (μmol/L), Hp (ng/mL) and APN (µg/mL). The variables that changed significantly were BMI (kg/m^2^), SBP (mmHg), DBP (mmHg), IL-6 (pg/mL), hsCRP (mg/L), APLN (ng/mL), RETN (ng/mL) and Lep (ng/mL) (Table 1).

The mean ± SD values for BMI in OW vs. NW (27.52 ± 1.42 vs. 21.49 ± 2.03) showed a highly significant difference (t: 33.89; df: 388; *p*: <0.0001). Serum Lep levels in OW vs. NW (10.70 ± 4.67 vs. 4.68 ± 1.91) indicated highly significant variation (t = 16.7476; df = 388; *p* < 0.0001) (Table 1). 

The SBP for OW vs. NW (121.37 ± 2.59 vs. 118.51 ± 1.54) was also found to be highly significant (t = 13.2859; df = 388; *p* < 0.0001) (Table 1). The values of DBP for OW vs. NW (81.44 ± 1.97 vs. 78.79 ± 1.44) gave a highly significant difference (t = 15.1955; df = 388; *p* < 0.0001) (Table 1).

### 3.1. Analysis of Variation in Serum Leptin and Other Variables in Normal-Weight and Overweight Subjects

The one-way ANOVA showed a significant variation in NW subjects for DBP, FBG, TC, IL-6, hsCRP, APN and Lep, and a significant variation in OW subjects for all variables, except Hp (Table 2).

The detailed analysis and comparison of the mean ± SD values obtained for various variables were carried out in NW and OW male university students (Table 3). The present data clearly show a significant increase in several variables at higher BMI levels. Most of the factors in the present study increased with an increase in BMI, except APN that showed a decreasing trend with an increasing BMI and Lep levels. 

A series of widespread significant variations were noticed in the levels of FBG and APN in NW subjects. Highly significant levels of serum Lep in the high–normal BMI population compared to those with low or medium–normal BMI levels were found in NW subjects (*p* < 0.001). The Lep levels increased with a significant change of quite high levels in OW subjects (*p* < 0.001; Table 3). 

Serum leptin levels showed significant variations at the high–normal level of BMI compared to the medium–normal and low normal levels of BMI in NW subjects. However, serum leptin in OW subjects showed significantly increased levels at various BMI levels of the high-overweight range (Table 3). The serum levels of various variables did not increase significantly in NW subjects at their low normal levels of BMI. Significant variation in SBP, TG, Hcy, Hp, APLN and RETN were not seen in NW subjects, though these variables changed significantly in OW subjects, especially at high overweight levels. Other variables, including Lep, DBP, TC, IL-6 and hsCRP, increased during high normal BMI levels; whereas, FBG, IL-6 and APN presented wide variations during the medium and high–normal level of BMI in NW subjects.

Higher levels of different factors, especially at higher and medium BMI levels, were observed in OW subjects. A highly significant increase in Lep, SBP, DBP, FBG, TC, TG, IL-6, hsCRP, Hcy, APN, APLN and RETN were obtained in OW subjects.

No significant difference could be found in Hp levels in both NW and OW subjects, though a significant change in IL-6, hsCRP and related factors was obtained.

Below, the given results in the present section are shown in Table 3.

The Tukey–Kramer test showed a significant difference in SBP for 25–25.9 vs. 27–27.9, 25–25.9 vs. 28–28.9, 25–25.9 vs. 29–29.9, 26–26.9 vs. 27–27.9, 26–26.9 vs. 29–29.9, 27–27.9 vs. 29–29.9 and 28–28.9 vs. 29–29.9 in OW subjects. The DBP in NW subjects gave a significant difference only for 18–18.9 vs. 24–24.9, whereas the DBP in OW subjects gave a significant difference for 25–25.9 vs. 27–27.9, 25–25.9 vs. 28–28.9, 25–25.9 vs. 29–29.9, 26–26.9 vs. 27–27.9, 26–26.9 vs. 28–28.9 and 26–26.9 vs. 29–29.9.

A significant difference of FBG in NW subjects was found in 18–18.9 vs. 24–24.9, 19–19.9 vs. 21–21.9, 19–19.9 vs. 22–22.9, 19–19.9 vs. 23–23.9 and 19–19.9 vs. 24–24.9. On the other hand, a significant difference in FBG for OW subjects was noted for 25–25.9 vs. 27–27.9, 25–25.9 vs. 28–28.9, 25–25.9 vs. 29–29.9, 26–26.9 vs. 28–28.9, 26–26.9 vs. 29–29.9, 27–27.9 vs. 29–29.9 and 28–28.9 vs. 29–29.9. TC was found to be significantly different for NW subjects only for 18–18.9 vs. 24–24.9, whereas significant variations in TC levels in OW subjects were noted for 25–25.9 vs. 27–27.9, 25–25.9 vs. 28–28.9, 25–25.9 vs. 29–29.9 and 26–26.9 vs. 29–29.9. The results showed a significant difference for TG in OW subjects for 25–25.9 vs. 27–27.9, 25–25.9 vs. 29–29.9, 26–26.9 vs. 29–29.9 and 28–28.9 vs. 29–29.9. 

The levels of IL-6 showed a significant difference in NW subjects for 18–18.9 vs. 23–23.9, 18–18.9 vs. 24–24.9, 19–19.9 vs. 23–23.9, 19–19.9 vs. 24–24.9, 20–20.9 vs. 23–23.9, 20–20.9 vs. 24–24.9 and 21–21.9 vs. 24–24.9. The results for IL-6 showed a significant difference in OW subjects for 25–25.9 vs. 28–28.9, 25–25.9 vs. 29–29.9, 26–26.9 vs. 28–28.9, 26–26.9 vs. 29–29.9, 27–27.9 vs. 29–29.9 and 28–28.9 vs. 29–29.9. The hsCRP had significant variation in NW subjects for 18–18.9 vs. 24–24.9, 19–19.9 vs. 24–24.9, 20–20.9 vs. 24–24.9 and 21–21.9 vs. 24–24.9, and in OW subjects for 25–25.9 vs. 29–29.9 and 26–26.9 vs. 29–29.9. No significant variations were obtained for Hcy in NW subjects for all comparisons. However, a significant difference was obtained in OW subjects for 25–25.9 vs. 29–29.9 and 26–26.9 vs. 29–29.9. The Hp in NW, as well as OW subjects for all comparisons did not present any significant change.

Significant variation was obtained for APN in NW subjects for 18–18.9 vs. 21–21.9, 18–18.9 vs. 22–22.9, 18–18.9 vs. 24–24.9, 19–19.9 vs. 22–22.9, 19–19.9 vs. 23–23.9 and 19–19.9 vs. 24–24.9, whereas a significant difference was obtained for APN in OW subjects for 25–25.9 vs. 27–27.9, 25–25.9 vs. 27–27.9, 25–25.9 vs. 29–29.9, 26–26.9 vs. 27–27.9, 26–26.9 vs. 28–28.9 and 26–26.9 vs. 29–29.9. The APLN in NW subjects for all comparisons gave non-significant variation, but a significant difference was obtained for APLN in OW subjects for 25–25.9 vs. 27–27.9 and 25–25.9 vs. 29–29.9. There was no significant variation for RETN in NW subjects for all comparisons, but a significant difference was obtained for APLN in OW subjects for 25–25.9 vs. 28–28.9 and 25–25.9 vs. 29–29.9.

Results showed significant difference for Lep in NW subjects for 18–18.9 vs. 23–23.9 and 18–18.9 vs. 24–24.9, whereas a significant difference was obtained for Lep in OW subjects for most of the comparisons (25–25.9 vs. 27–27.9, 25–25.9 vs. 28–28.9, 25–25.9 vs. 29–29.9, 26–26.6 vs. 27–27.9, 26–26.6 vs. 28–28.9, 26–26.9 vs. 29–29.9 and 27–27.9 vs. 29–29.9).

### 3.2. Association of the Body Mass Index and Serum Leptin in Normal-Weight and Overweight Subjects

BMI was plotted against serum Lep (Table 4). The R^2^ value was found to be 0.14 for NW (BMI range: 18–24.9) and 0.34 for OW (BMI range: 25–29.9), that showed highly significant (*p* < 0.0001) positive linear correlations. The subgroups of BMI showed a significant correlation with Lep only at BMI 20–20.9 in NW, but showed a highly significant positive association in most of the plots in OW (Table 4). 

### 3.3. Association of Body Mass Index and Systolic Blood Pressure in Normal-Weight and Overweight Subjects

The BMI and SBP plotted against each other (Table 4) showed the R^2^ value for NW to be 0.01 (*p*> 0.05) and OW 0.34 (*p* < 0.0001). This indicated a positive linear and highly significant linear correlation for OW, but a nonsignificant relationship for NW subjects. The subgroups of BMI did not show any significant correlation with SBP in NW, but showed a highly significant positive association in most of the plots in OW (Table 4). 

### 3.4. Association of Body Mass Index and Diastolic Blood Pressure in Normal-Weight and Overweight Subjects

The plot between BMI and DBP (Table 4) showed the R^2^ value 0.08 for NW and 0.31 for OW subjects. A positive linear and highly significant relationship was found present between BMI and DBP in both groups of subjects. The subgroups of BMI did not show any significant correlation with DBP in NW, but showed a highly significant positive association in most of the plots in OW (Table 4). 

### 3.5. Association of Body Mass Index with Other Variables in Normal-Weight and Overweight Subjects

The results of the present section are given below, are shown in Table 4.

The FBG indicated significant correlation with BMI at BMI levels 19–19.9, 22–22,9 and 23–23.9 and for whole data of NW subjects. Whereas the BMI levels 26–26.9 and 29–29.9 and whole data of OW presented a significant linear correlation. The TC showed a significant correlation with BMI in all BMI levels in NW and in the last level of BMI (29–29.9) and whole data record (BMI: 25–29.9) in OW subjects. The TG gave a significant positive linear correlation with most of the BMI levels in NW subjects, whereas in only last level studied in OW subjects. The whole data of NW and OW subjects showed highly significant correlations.

Serum IL-6 presented a highly significant correlation for both NW and OW data for whole data. However, a significant correlation with most of the higher levels of BMI with IL-6 was obtained in NW subjects. The OW subjects showed a significant association in the BMI level of 29–29.9. The hsCRP presented a significant correlation in almost all BMI subgroups in NW subjects and in the higher level of 29–29.9 in OW subjects. However, the whole data of both NW and OW subjects showed highly significant correlations. The BMI range value of 18–24.9 in NW and 25–29.9 in OW subjects showed highly significant correlations with Hcy. Higher levels of BMI in NW subjects also gave a significant correlation. The lower and higher levels of BMI showed significant correlations with Hp. The BMI levels of 18–24.9 and 25–29.5 gave significant correlations, respectively, for NW and OW subjects.

The initial level of BMI (18–18.9) in NW subjects and whole data altogether for NW and OW subjects showed significant correlations with the APN. A highly significant correlation between BMI and APLN was obtained in the whole data (BMI: 25–29.9), as well as in several BMI levels in OW subjects, whereas NW subjects only showed a significant correlation in whole data (BMI: 18–24.9). A significant correlation that was too high was noticed between BMI and APLN at all BMI range levels in NW subjects and in the BMI level of 25–25.9 in OW subjects. Furthermore, both complete-range BMI levels in NW and OW subjects gave highly significant correlations.

### 3.6. Association of Serum Leptin and Systolic Blood Pressure in Normal-Weight and Overweight Subjects

The SBP and Serum Lep levels were plotted (Table 5). The plot showed the R^2^ value 0.26 for NW and 0.63 for OW subjects. A positive linear and significant relationship was found present between SBP and serum Lep in both subject groups (Table 5). The correlation of Lep and SBP in the BMI subgroups showed significant correlation for all plots in NW and OW, except at the 28–29.9 BMI level (Table 5). 

### 3.7. Association of Serum Leptin and Diastolic Blood Pressure in Normal-Weight and Overweight Subjects

The DBP and serum Lep levels were plotted (Table 5). The plot showed an R^2^ value of 0.29 for NW and 0.45 for OW subjects, indicating a highly significant positive linear relationship between DBP and serum Lep in both subject groups. The association of Lep and DBP showed a significant positive association in all plots in NW and most of the plots in OW subjects (Table 5). 

### 3.8. Association of Serum Leptin with Other Variables in Normal-Weight and Overweight Subjects

Below, the given results in the present section are shown in Table 5.

The plot of serum Lep and FBG showed a significant correlation only in OW subjects at BMI level 26–26.9 and whole data (BMI 25–29.9). Serum Lep and TC correlated significantly, mainly for whole BMI data for both NW and OW subjects. Serum Lep and TG correlated significantly in the higher level of BMI (29–29.9) and whole data of OW subjects.

The Lep and IL-6 correlated significantly only with the whole data of OW subjects. The Lep and hsCRP correlated with a high significance with the whole data of OW subjects at a higher level of BMI (29–29.9), as well as in the whole data of subjects with NW. The correlation for Lep and Hcy was highly significant only with the whole data of OW subjects. The correlations for Lep and Hp were found at one high level of BMI in NW and one higher level of BMI in OW subjects. The whole data of both NW and OW subjects showed highly significant correlations.

Correlations for Lep and APN were found at a medium BMI level (21–21.9) in NW and at higher levels of BMI (28–28.9 and 29–29.9) in OW subjects. The correlations for Lep and APLN were highly interesting. They correlated significantly at the lower and higher levels of BMI with considerable progressive pattern for both NW and OW subjects. Highly significant correlations were obtained with the whole BMI-based data in NW and OW subjects. The correlations for Lep and RETN showed a highly significant association with the whole BMI-based data of NW and OW subjects.

### 3.9. Association of Systolic Blood Pressure with Other Variables in Normal-Weight and Overweight Subjects

Below, the given results in the present section are shown in Table 6.

The SBP and DBP correlated with high significance at all BMI levels and whole data BMI levels in NW and OW subjects. The SBP was associated with a high significance with FBG only among OW subjects having a BMI of 29–29.9 and whole data BMI. All other plots for NW and OW subjects showed non-significant associations. The association of SBP and TC was found to be highly significant at whole data BMI in OW subjects. All other plots gave non-significant results. The correlation of SBP and TG was also found to be highly significant at whole data BMI in OW subjects. All other plots gave non-significant results.

The association of SBP and IL-6 was found to be highly significant for whole data BMI in OW subjects. All other results were non-significant. The plot of SBP against hsCRP presented highly significant association in NW and OW subjects besides the initial BMI level (18–18.9) in NW subjects and a higher BMI level (29–29.9) in OW subjects. The SBP was significantly associated with Hcy at a lower and a higher BMI level in NW subjects, and highly significant with the whole data BMI in OW subjects. The SBP association with Hp was found significant at BMI 25–29.9 in OW subjects and BMI 18–24.9 in NW subjects. A medium and a higher level of BMI in OW subjects, and two medium BMIs in NW subjects are shown in Table 6, displaying the significant correlation with Hp.

The whole data of BMI and about a half of the BMI sub-levels showed a significant association of SBP with APN in OW subjects. Two medium sub-levels of BMI in NW subjects also presented a significant association. Highly interesting results were obtained for the association between SBP and APLN for both NW and OW subjects. Both groups of subjects for whole data BMI and all sub-levels of BMI showed highly significant correlations. A highly significant correlation between SBP and RETN was obtained at the whole data BMI in OW subjects. Another significant association was found at BMI sub-level of 26–26.9. 

### 3.10. Association of Diastolic Blood Pressure with Other Variables in Normal-Weight and Overweight Subjects

Below, the given results in the present section are shown in Table 7.

The plot of DBP against FBG showed a highly significant association for BMI 29–29.9 and whole data BMI level in OW subjects. The NW subjects did not show any significant results. The plot for DBP and TC gave a significant association only for BMI 18–24.9 in NW and 25–29.9 in OW subjects. The DBP and TG were plotted showing only a significant correlation for BMI 25–29.9 in OW subjects.

The association of DBP and IL-6 was found to be highly significant only for BMI 25–29.9 in OW subjects. The correlation of DBP against hsCRP was significant mainly at BMI levels of 18–24.9 in NW subjects and 25–29.9 in OW subjects. One other significant result for the initial BMI level (18–18.9) in NW subjects and one for a higher BMI level (29–29.9) in OW subjects was obtained. The results for DBP against Hcy presented a significant association at BMI levels of 25–29.9 and 29–29.9 in OW subjects, and a lower (18–18.9) and a higher (23–23.9) BMI in NW subjects. The correlation for DBP and Hp gave a highly significant association for BMI 25–29.9 in OW subjects. Other higher BMI levels showing a significant association were 29–29.5 and 27–27.9 in OW subjects, and for lower (20–20.9) and medium (2–22.9) BMI levels.

Significant results for the correlation between DBP and APN were found for BMI 26–26.9 in OW subjects and 20–20.9 in NW subjects. The most interesting results were the highly significant correlations of DBP against APLN in NW, as well as OW subjects for all BMI sub-levels. The RETN against DBP presented a highly significant association at BMI 25–29.9 and BMI 26–26.9 in OW subjects.

## 4. Discussion

The present study provides information about the significant associations among BP (SBP and DBP), BMI and serum leptin levels in young NW and OW male Saudi students. These findings are quite interesting in general, especially for the quality of the healthy life of the Saudi population. An association of the characteristics and other variables of the subjects in the present report relate to the interactive role of various variables during the progressive increase in BMI. Reports of the impact of inflammation involving IL-6 [37,38], CRP [39], APN [38,39,40], APLN [41] and RETN [37] in association with Lep could further explain the association among BMI, blood pressure and leptin levels in NW, OW and obese subjects. The most interesting results in the present study were for serum APLN that significantly correlated with Lep, BMI, SBP and DBP in lower and higher levels of BMI with a considerable progressive pattern in both NW and OW subjects. 

In the present study, the elevated levels of serum Lep in overweight male subjects with an elevated BMI can be interpreted by a previous study investigating the independent association of elevated Lep with the greater BMI in overweight children and adolescents [17]. The positive correlation of Lep levels and BMI was revealed [15,16,17,18]. It was found that serum Lep was significantly associated with BMI [20,42]. Another report reveals that the Lep serum levels increase with the increase in BMI toward obesity, manifesting Lep resistance [43]. An increase in Lep levels was investigated in overweight subjects as compared to subjects with a normal weight, and it was found that there is a positive correlation between Lep and BMI [44]. Our present results orient toward the investigation that BMI is significantly associated with Lep, along with triglycerides, in both men and women, and explains the origin of Lep resistance [45]. The present results, showing a linear and significant association between serum Lep and BMI, are similar to those of previous investigations [17,18,19]. Furthermore, the results in the present study do not contradict previous studies [8,10,11,20]. 

The serum Lep and DBP, with a significant linear association in the present report, resemble another previous study [21], though the absence of any significant correlation of Lep with SBP or DBP was investigated [16] in subjects of another age group. A later report showed that serum Lep is positively correlated with DBP and TC [42].

In the present study, the association of BP and serum Lep levels [23] and the significant association between serum leptin and SBP for NW and OW subjects are also in line with results from several other studies [8,10,15,24,25,26]. A positive linear and significant association between serum Lep and DBP partly resembles the findings in a report for subjects where Lep was positively and significantly associated with DBP, but not with SBP [21]. Elevated levels of serum Lep, SBP and DBP in overweight male subjects, and a positive linear correlation, especially in subjects with a high normal weight and high overweight status, is evident in the reports, presenting association of Lep levels with BP [8,10,15], only SBP [15] or only DBP [21], along with the controversial association of Lep with BMI and BP [6,10].

Our results, presenting elevated levels of serum Lep and BP at higher BMI levels, are similar to the investigation that an increasing BP increases the BMI, and hence serum Lep in male adolescents, manifesting a significant association among BMI, SBP and the mean BP [15]. Studies reveal that Lep is an independent factor for obesity associated an elevation in BP, and Lep is especially associated at higher levels of BP (SBP and DBP) and BMI [19,24,25]. The interactive activity of an elevated BMI and elevated BP in association with increased Lep in obesity-related hypertension in adolescents is quite evident in another report where BP was classified into categories of normal, high–normal and hypertensive, even after adjusting for the involvement of BMI and age [26].

In the present study, some BMI levels, especially in subjects with a lower normal weight and lower overweight, did not show the significant variations or correlations for Lep and SBP, and Lep and DBP. Similar findings were documented previously [16]. It was explained that Lep may increase DBP in obesity, being only a mediator, and even without an association between Lep and mean SBP or DBP in subjects with high BP [21], as we obtained similar results at certain levels of BMI in overweight subjects. We found significant positive linear correlations of Lep with SBP or DBP, even in normal-weight subjects, but not in overweight subjects. Our results for Lep, SBP, DBP and various other factors at individual BMI levels differed from whole data BMI levels in normal-weight and overweight subjects. This explains the previous findings [16,21].

Our present work shows the association of Lep with SBP, as well as DBP in male university students, though it was found that Lep associates with DBP, but not the SBP [21], and Lep associates with BMI, but not with BP [27]. The associations of Lep, BP and BMI are quite complicated. This association may vary with age, sex, race and type of obesity [11].

For the association of Lep and FBG, the present results show a highly significant correlation in OW subjects, similar to another study where serum Lep was significantly associated with FBG [20]. Leptin has glucose-lowering effects independent of the regulation of Lep for body weight [46]. Hypothalamic glucose-sensing for the rise or fall of plasma glucose levels has a prominent role in the regulation of glucose control in the brain by Lep for non-obese and obese subjects [47]. Serum Lep and TC were associated at a high significance with NW and OW subjects in the present study. The mean Lep was found to have a significant positive correlation with body weight, BMI, TC and TG [44]. The BMI, TC, TG, APN and RETN showed significant variations between the subjects having abnormal Lep serum levels and those having normal Lep serum levels [48]. The mean Lep was found to have a significant positive correlation with TG [44]. Our results show a more profound correlation in OW subjects. The BMI and TG showed a significant variation between the subjects having abnormal Lep serum levels and those having normal Lep serum levels [48].

A significant association between Lep and APN (without significant variation for NW vs. OW), mainly at higher BMI sub-levels, was obtained in the current report. In another report, the BMI and APN showed significant variation between the subjects having abnormal Lep serum levels and those having normal Lep serum levels [48]. Variations of Lep and RETN for NW vs. OW subjects and the correlation at higher BMI levels and whole data BMI showed highly significant results in our present study. Another study revealed that the BMI and RETN had significant variation between the subjects having abnormal Lep serum levels and those having normal Lep serum levels [48].

A significant difference in our work was obtained for inflammatory cytokine IL-6 for NW and OW subjects, and a significant association for Lep and IL-6 was obtained for NW and OW subjects. Another study showed an association of moderate-level serum Lep and the IL-6 [48]. The serum levels of hs-CRP were found to be positively associated with Lep, and independently associated with BMI [49]. Our results show a highly significant association between Lep and hsCRP in both NW and OW subjects. The Lep was found to be highly associated with hsCRP, and it was suggested to improve body weight for reversing the obesity-associated chronic state of inflammation [50].

Our study presented non-significant variation of Hcy for NW vs. OW subjects. Obese patients in another report showed significantly higher Hcy, glucose, Lep, TG and hsCRP levels than the controls [51]. We found a highly significant linear correlation in OW subjects for Lep against Hcy. hsCRP, Hp and Lep were found at higher levels, APN was unchanged and no statistical difference for IL-6 was found in obese compared to healthy subjects (Sal et al., 2018) [52]. The serum levels of Hp were non-significantly different in NW vs. OW subjects, though the association of Lep and Hp was highly significant in both NW and OW subjects.

The present study demonstrated quite informative results regarding serum APLN that correlated significantly with Lep, BMI, SBP and DBP at lower and higher levels of BMI, with considerable progressive patterns in both NW and OW subjects. The APLN levels in the present study showed a highly significant correlation with SBP and DBP for both groups (NW and OW) and all BMI subgroups, and an association of Lep and APLN presented a highly significant correlation. In hypertensive overweight/obese children, Lep and APLN levels were found to be statistically higher than normal controls and NW subjects with hypertension [53].

We found that the levels of various factors varied significantly at lower and higher levels of the body mass index in normal-weight and overweight male students. However, further studies may clarify the variations and associations of variables, even among the normal-weight and overweight status, by describing them as high–normal weight and high overweight status. In general, the serum leptin levels in the male university students studied presently were significantly higher in the overweight group as compared to the normal-weight group, with a concomitant significant increase in apelin, resistin, SBP, DBP, IL-6 and hsCRP. It was further found that a significant positive linear correlation of serum leptin was present in OW subjects (BMI: 25–29.9 kg/m^2^) with apelin, resistin, SBP, DBP, Il-6, hsCRP and other variables, including FBG, TC, TG, Hcy and Hp. The mentioned variations and associations provide a background for elucidating the underlying mechanism explaining the steady elevation in serum leptin, along with other related factors, especially in high OW subjects. 

It is known that increased level of circulating leptin is an important biomarker of leptin resistance that is quite common in obese people. Hence, our observations in the present study led us to suggest that the focus of further studies should be identifying new mechanisms of leptin regulation at the whole-body level, to design new novel drug products that reverse leptin resistance. In this regard, understanding the pathogenesis of overweight-related and obesity-related disorders, and the control of energy homeostasis by leptin may provide new alternatives for overweight/obesity treatment. Pharmaceutical companies could further pursue the idea of employing leptin-based drug products as a therapeutic strategy for achieving weight loss. There is a strong possibility of using leptin as a treatment for overweight and high overweight, obesity and OW/obesity-related disorders. However, it is important to carry out precise studies in OW and obese men and women with a wide age range to understand the physiology and pathophysiology of leptin-associated mechanisms for OW, high OW and obesity status populations, leptin resistance/related disorders and the biological/clinical importance of leptin for new strategies in novel therapeutic approaches. 

## 5. Limitations

Our pilot studies indicated a wide difference in the levels of leptin and other factors, while considering age and gender. Hence, we attempted to initiate from a narrow age range of only male students, though there were also financial reasons for limiting our study to a narrow age range and gender. The project funded for the present research by the Umm Al-Qura University was solely for the students entering the university education (initial two to three years; around 18–20 years age). 

There were several other limitations in our study. The samples were collected of only normal-weight and overweight subjects, and not for obese subjects. Further work may clarify the associations among BMI, BP and leptin levels in obese subjects. The present study was carried out in subjects with an age range of 18–20 years. However, specifying a narrow age range and gender are the study’s limitations. To make up for the deficiency, we have recently started studying male and female students with a wider age range. Findings with a lower and higher age range (than the current 18–20 years) are required to be explored in both male and female subjects, to have a better idea of the precise age- and gender-based changes in BP and serum leptin levels under the influence of changing BMI levels.

## Figures and Tables

**Table 1 jpm-13-00828-t001:** Characteristics and other variables of normal-weight and overweight study subjects.

Variable	Normal-Weight and Overweight Subjects		*p*-Value
	NW	OW	
Number of subjects (n)	198	192	-
Sex (male)	198	192	-
Age (years)	18.88 ± 0.67	18.83 ± 0.70	NS
BMI (kg/m^2^)	21.49 ± 2.03	27.52 ± 1.42	<0.0001
SBP (mmHg)	118.51 ± 1.54	121.37 ± 2.59	<0.0001
DBP (mmHg)	78.79 ± 1.44	81.44 ± 1.97	<0.0001
FBG (mg/dL)	91.26 ± 1.78	91.56 ± 1.34	NS
Total cholesterol (mg/dL)	170.25 ± 6.34	171.48 ± 7.51	NS
Triglycerides (mg/dL)	91.35 ± 5.71	92.74 ± 9.36	NS
IL-6 (pg/mL)	3.41 ± 2.63	3.98 ± 2.78	<0.04
hsCRP (mg/L)	0.88 ± 0.45	0.98 ± 0.51	<0.04
Hcy (μmol/L)	4.92 ± 2.87	4.98 ± 2.93	NS
Hp (ng/mL)	9.87 ± 5.27	9.78 ± 7.20	NS
Adiponectin (µg/mL)	5.53 ± 3.03	5.48 ± 2.98	NS
Apelin (ng/mL)	2.30 ± 1.58	2.76 ± 1.97	<0.01
Resistin (ng/mL)	6.33 ± 2.38	7.23 ± 3.17	<0.002
Leptin (ng/mL)	4.68 ± 1.91	10.70 ± 4.67	<0.0001

NW: normal weight; OW: overweight; BMI: body mass index; SBP: systolic blood pressure; DBP: diastolic blood pressure; FBG: fasting blood glucose; IL-6: interleukin-6; hsCRP: high-sensitivity C-reactive protein; Hcy: homocysteine; Hp: hepcidin; values are the mean ± standard deviation (SD); a two-tailed *p*-value was obtained by applying an unpaired two-sample *t*-test.

**Table 2 jpm-13-00828-t002:** Analysis of variance on the basis of various levels of the body mass index in normal-weight and overweight male university students.

Variable	ANOVA			
	NW Subjects (n:198) at Various BMI Levels		OW Subjects (n: 192) at Various BMI Levels	
	F	*p*	F	*p*
SBP (mmHg)	0.52	0.794	21.44	<0.0001
DBP (mmHg)	2.90	0.010	16.56	<0.0001
FBG (mg/dL)	5.21	<0.0001	14.67	<0.0001
Total cholesterol (mg/dL)	2.91	0.0098	08.21	<0.0001
Triglycerides (mg/dL)	2.01	0.067	08.05	<0.0001
IL-6 ((pg/mL)	6.92	<0.0001	12.33	<0.0001
hsCRP (mg/L)	6.31	<0.0001	06.50	<0.0001
Hcy (μmol/L)	1.68	0.129	03.96	0.004
Hp (ng/mL)	0.58	0.749	01.08	0.369
Adiponectin (µg/mL)	7.47	<0.0001	18.10	<0.0001
Apelin (ng/mL)	1.49	0.182	05.91	0.0001
Resistin (ng/mL)	0.33	0.922	04.38	0.002
Leptin (ng/mL)	4.83	0.0001	18.23	<0.0001

ANOVA: analysis of variance; NW: normal weight; OW: overweight; BMI: body mass index; SBP: systolic blood pressure; DBP: diastolic blood pressure; FBG: fasting blood glucose; IL-6: interleukin-6; hsCRP: high-sensitivity C-reactive protein; Hcy: homocysteine; Hp: hepcidin; values are the mean ± standard deviation (SD); ANOVA for SBP in NW subjects at BMI levels of 18–18.9, 19–19.9, 20–20.9, 21–21.9, 22–22.9, 23–23.9 and 24–24.9 kg/m^2^; ANOVA for SBP in OW subjects at BMI levels of 25–25.9, 26–26.9, 27–27.9, 28–28.9 and 29–29.9; one-way ANOVA was obtained by using the Statistical Package for Social Sciences (SPSS), version 24.0 for Windows.

**Table 3 jpm-13-00828-t003:** Comparison of the levels of various variables in normal-weight and overweight male university students.

Variable	BMI-Based Mean± SD Levels and *p*-Values for Various Variables											
	BMI of NW Subjects (n: 198)							BMI of OW Subjects (n: 192)				
	18–18.9 (n: 29)	19–19.9 (n: 29)	20–20.9 (n: 28)	21–21.9 (n: 27)	22–22.9 (n: 29)	23–23.9 (n: 28)	24–24.9 (n: 28)	25–25.9 (n: 38)	26–26.9 (n: 38)	27–27.9 (n: 40)	28–28.9 (n: 37)	29–29.9 (n: 39)
SBP (mmHg)	118.31 ± 1.58	118.55 ± 1.55	118.39 ± 1.47	118.56 ± 1.50	118.66 ± 1.67	118.25 ± 1.62	118.86 ± 1.46	119.26 ± 1.25	120.50 ± 1.35	121.95 ± 1.89 O2 *** O5 ***	121.46 ± 1.61 O3 ***	123.59 ± 3.70 O4 *** O7 *** O9 * O10 **
DBP (mmHg)	78.21 ± 1.66	78.55 ± 1.43	78.54 ± 1.43	78.48 ± 1.22	79.24 ± 1.27	79.14 ± 1.30	79.36 ± 1.47 N6 **	79.87 ± 1.85	80.74 ± 1.29	81.88 ± 1.64 O2 *** O5 **	81.97 ±1.61 O3 *** O6 ***	82.72 ± 2.08 O4 *** O7 ***
FBG (mg/dL)	90.75 ± 1.99	90.02 ± 1.47	91.08 ± 1.44	91.48 ± 1.50 N8 ***	91.52 ± 1.74 N9 ***	91.84 ± 1.46 N10 ***	92.16 ± 2.02 N6 * N11 ***	90.75 ± 0.86	90.97 ± 0.96	91.65 ± 1.35 O2 ***	91.80 ± 1.16 O3 *** O6 **	92.60 ± 1.48 O4 *** O7 *** O9 ** O10 **
TC (mg/dL)	167.10 ± 6.02	169.00 ± 6.93	169.75 ± 6.76	169.96 ± 6.58	171.41 ± 6.46	171.61 ± 5.29	173.07 ± 4.43 N6 ***	167.16 ± 7.42	170.04 ± 6.34	172.14 ± 7.94 O2 **	172.00 ± 6.89 O3 **	175.99 ± 6.22 O4 *** O7 ***
TG (mg/dL)	90.24 ± 5.71	90.14 ± 5.31	90.25 ± 5.52	91.00 ± 5.02	91.52 ± 5.69	92.14 ± 5.89	94.32 ± 6.16	88.26 ± 8.13	90.21 ± 5.95	94.13 ± 6.96 O2 ***	92.03 ± 7.60	98.64 ± 13.14 O4 *** O7 *** O10 **
IL-6 (pg/mL)	2.21 ± 0.94	2.23 ± 1.35	2.67 ± 1.66	3.04 ± 1.63	3.94 ± 2.49	4.67 ± 3.08 N5 *** N10 *** N14 **	5.22 ± 4.20 N6 *** N11 *** N15 ** N18 *	2.71 ± 1.77	2.79 ± 1.56	3.82 ± 2.24	4.38 ± 2.48 O3 ** O6 **	6.16 ± 3.78 O4 *** O7 *** O9 ** O10 ***
hsCRP (mg/L)	0.72 ± 0.31	0.69 ± 0.30	0.75 ± 0.39	0.81 ± 0.40	0.95 ± 0.42	0.98 ± 0.40	1.26 ± 0.65 N6 *** N11 *** N15 *** N18 **	0.84 ± 0.28	0.76 ± 0.28	0.99 ± 0.56	0.99 ± 0.51	1.29 ± 0.68 O4 *** O7 ***
Hcy (μmol/L)	4.23 ± 0.51	4.49 ± 1.45	4.42 ± 1.22	4.88 ± 1.42	4.87 ± 2.06	5.46 ± 2.99	6.45 ± 6.22	4.22 ± 0.54	4.47 ± 1.40	4.83 ± 1.63	4.82 ± 1.98	6.54 ± 5.61 O4 *, O7 *
Hp (ng/mL)	9.49 ± 4.61	8.79 ± 4.63	9.72 ± 5.05	9.67 ± 5.19	9.67 ± 5.57	10.84 ± 5.40	10.91 ± 6.55	8.79 ± 6.60	8.92 ± 5.16	9.75 ± 5.95	9.51 ± 6.87	11.80 ± 10.48
APN (µg/mL)	7.97 ± 4.50	6.80 ± 3.09	5.82 ± 2.66	4.78 ± 2.36 N3 **	4.56 ± 1.96 N4 *** N9 **	4.36 ± 1.95 N5 *** N10 ***	4.29 ± 1.90 N6 *** N11 ***	7.80 ± 3.48	6.86 ± 3.51	4.80 ± 1.89 O2 *** O5 **	4.41 ± 1.40 O3 *** O6 ***	3.62 ± 1.68 O4 *** O7 ***
APLN (ng/mL)	1.86 ± 1.22	1.96 ± 1.11	2.31 ± 2.12	2.13 ± 0.93	2.91 ± 2.65	2.33 ± 0.96	2.52 ± 1.01	1.68 ± 0.82	2.57 ± 0.85	2.94 ± 0.44 O2 ***	2.84 ± 0.47	3.73 ± 3.96 O4 **
RETN (ng/mL)	6.29 ± 2.30	6.28 ± 1.58	6.16 ± 2.37	6.16 ± 2.37	6.49 ± 2.28	6.55 ± 2.49	6.61 ± 3.06	6.14 ± 2.38	6.76 ± 2.75	6.60 ± 2.94	8.42 ± 3.14 O3 ***	8.28 ± 3.91 O4 **
Lep (ng/mL)	3.58 ± 1.73	4.31 ± 1.55	4.49 ± 1.76	4.36 ± 1.67	4.78 ± 1.86	5.72 ± 1.68 N5 ***	5.56 ± 2.34 N6 ***	7.40 ± 1.81	8.48 ± 2.40	11.20 ± 2.71 O2 *** O5 ***	12.15 ± 3.88 O3 *** O6 ***	14.20 ± 6.95 O4 *** O7 *** O9 *

NW: normal weight; OW: overweight; BMI: body mass index; SBP: systolic blood pressure; DBP: diastolic blood pressure; FBG: fasting blood glucose; TC: total cholesterol; TG: triglycerides; IL-6: interleukin-6; hsCRP: high-sensitivity C-reactive protein; Hcy: homocysteine; Hp: hepcidin; APN: adiponectin; APLN: Apelin; RETN: resistin; Lep: leptin; values are the mean± standard deviation (SD); a two-tailed *p*-value was obtained by applying an unpaired two-sample *t*-test; *: *p* < 0.05; **: *p* < 0.01; ***: *p* < 0.001. Various BMI comparisons for NW subjects represented as: N1:18–18.9 vs.19–19.9, N2:18–18.9 vs. 20–20.9, N3:18–18.9 vs. 21–21.9, N4:18–18.9 vs. 22–22.9, N5:18–18.9 vs. 23–23.9, N6:18–18.9 vs. 24–24.9, N7:19–19.9 vs. 20–20.9, N8:19–19.9 vs. 21–21.9, N9:19–19.9 vs. 22–22.9, N10:19–19.9 vs. 23–23.9, N11:19–19.9 vs. 24–24.9, N11:19–19.9 vs. 24–24.9, N12:20–20.9 vs. 21–21.9, N13:20–20.9 vs. 22–22.9, N14:20–20.9 vs. 23–23.9, N15:20–20.9 vs. 24–24.9, N16:21–21.9 vs. 22–22.9, N17:21–21.9 vs. 23–23.9, N18:21–21.9 vs. 24–24.9, N19:22–22.9 vs. 23–23.9, N20:22–22.9 vs. 24–24.9, N21:23–23.9 vs. 24–24.9. Various BMI comparisons for OW subjects represented as: O1:25–25.9 vs. 26–26.9, O2:25–25.9 vs. 27–27.9, O3:25–25.9 vs. 28–28.9, O4:25–25.9 vs. 29–29.9, O5:26–26.9 vs. 27–27.9, O6:26–26.9 vs. 28–28.9, O7:26–26.9 vs. 29–29.9, O8:27–27.9 vs. 28–28.9, O9:27–27.9 vs. 29–29.9, O10:28–28.9 vs. 29–29.9.

**Table 4 jpm-13-00828-t004:** Correlation of the body mass index with other variables in normal-weight and overweight male university students.

Variable	Coefficient of Determination for the Correlation of Body Mass Index with Serum Leptin, Blood Pressure and Other Variables													
	BMI of NW Subjects (n: 198)								BMI of OW Subjects (n: 192)					
	18–18.9	19–19.9	20–20.9	21–21.9	22–22.9	23–23.9	24–24.9	18–24.9	25–25.9	26–26.9	27–27.9	28–28.9	29–29.9	25–29.9
SBP (mmHg)	0.08	0.03	0.01	0.00	0.05	0.07	0.04	0.01	0.46 ***	0.07	0.26 ***	0.03	0.26 ***	0.34 ***
DBP (mmHg)	0.05	0.04	0.03	0.01	0.08	0.00	0.01	0.08 ***	0.52 ***	0.01	0.36 ***	0.00	0.22 **	0.31 ***
FBG (mg/dL)	0.11	0.14 *	0.06	0.09	0.26 **	0.29 **	0.32 **	0.15 **	0.01	0.11 *	0.07	0.04	0.23 **	0.27 **
TC (mg/dL)	0.33 **	0.73 ***	0.82 ***	0.74 ***	0.66 ***	0.65 ***	0.56 ***	0.15 ***	0.06	0.01	0.00	0.02	0.08	0.15 ***
TG (mg/dL)	0.49 ***	0.31 **	0.27 **	0.07	0.44 ***	0.31 **	0.45 ***	0.09 ***	0.04	0.02	0.00	0.01	0.19 **	0.14 ***
IL-6 (pg/mL)	0.05	0.11	0.23 **	0.13	0.25 **	0.26 **	0.32 **	0.21 ***	0.03	0.00	0.02	0.05	0.09	0.21 ***
hsCRP (mg/L)	0.07	0.14 *	0.25 **	0.18 *	0.38 ***	0.22 *	0.16 *	0.18 ***	0.03	0.00	0.01	0.06	0.23 **	0.12 ***
Hcy (μmol/L)	0.02	0.13	0.08	0.14	0.17 *	0.23 **	0.20 *	0.07 ***	0.01	0.00	0.03	0.02	0.10	0.07 ***
Hp (ng/mL)	0.06	0.25 **	0.09	0.20 *	0.14	0.09	0.14	0.03 *	0.02 *	0.03 *	0.08	0.11	0.07	0.03 *
APN (µg/mL)	0.20 *	0.00	0.03	0.03	0.08	0.04	0.14 *	0.14 ***	0.00	0.02	0.01	0.09	0.04	0.24 ***
APLN (ng/mL)	0.04	0.00	0.00	0.02	0.11	0.05	0.02	0.023 *	0.12 *	0.06	0.13 *	0.04	0.15 *	0.12 ***
RETN (ng/mL)	0.76 ***	0.34 ***	0.63 ***	0.38 ***	0.30 **	0.21 *	0.35 ***	0.02 *	0.24 **	0.01	0.03	0.01	0.05	0.09 ***
Lep (ng/mL)	0.10	0.04	0.23 **	0.13	0.08	0.03	0.00	0.14 ***	0.36 ***	0.33 ***	0.22 **	0.07	0.22 **	0.34 ***

NW: normal weight; OW: overweight; BMI: body mass index; SBP: systolic blood pressure; DBP: diastolic blood pressure; FBG: fasting blood glucose; TC: total cholesterol; TG: triglycerides; IL-6: interleukin-6; hsCRP: high-sensitivity C-reactive protein; Hcy: homocysteine; Hp: hepcidin; APN: adiponectin; APLN: Apelin; RETN: resistin; Lep: leptin; values are the mean ± standard deviation (SD); values are shown as the coefficient of determination (R^2^); *: *p* < 0.05; **: *p* < 0.01; ***: *p* < 0.001; regression lines were plotted for obtaining the values of R^2^ and the values of significance (*p*) using the Statistical Package for Social Sciences (SPSS), version 24.0 for Windows.

**Table 5 jpm-13-00828-t005:** Correlation of serum leptin with other variables on the basis of the body mass index in normal-weight and overweight male university students.

Variable	Coefficient of Determination for Correlation of Serum Leptin with Other Variables													
	BMI of NW Subjects (n: 198)								BMI of OW Subjects (n: 192)					
	18–18.9	19–19.9	20–20.9	21–21.9	22–22.9	23–23.9	24–24.9	18–24.9	25–25.9	26–26.9	27–27.9	28–28.9	29–29.9	25–29.9
SBP (mmHg)	0.33 **	0.33 **	0.21 *	0.38 ***	0.18 *	0.35 ***	0.38 ***	0.26 **	0.34 ***	0.22 **	0.62 ***	0.10	0.75 ***	0.63 ***
DBP (mmHg)	0.22 **	0.34 ***	0.28 **	0.47 ***	0.18 *	0.14 *	0.27 **	0.29 ***	0.30 ***	0.02	0.52 ***	0.09	0.56 ***	0.45 ***
FBG (mg/dL)	0.07	0.00	0.00	0.02	0.04	0.12	0.00	0.01	0.12 *	0.15 *	0.01	0.00	0.11 *	0.17 ***
TC (mg/dL)	0.06	0.02	0.15 *	0.05	0.01	0.05	0.02	0.06 ***	0.01	0.06	0.01	0.02	0.01	0.04 ***
TG (mg/dL)	0.02	0.02	0.17 *	0.099	0.08	0.01	0.081	0.01	0.02	0.05	0.00	0.00	0.10*	0.11 ***
IL-6 (pg/mL)	0.01	0.01	0.01	0.01	0.06	0.07	0.08	0.03 *	0.00	0.01	0.04	0.01	0.01	0.05 **
hsCRP (mg/L)	0.00	0.04	0.00	0.17 *	0.02	0.02	0.09	0.05 **	0.00	0.03	0.01	0.01	0.20 **	0.11 ***
Hcy (μmol/L)	0.04	0.10	0.01	0.00	0.01	0.01	0.04	0.00	0.09	0.02	0.05	0.00	0.05	0.06 ***
Hp (ng/mL)	0.00	0.00	0.09	0.24 **	0.12	0.01	0.00	0.03 **	0.00	0.00	0.13 *	0.00	0.12 *	0.07 ***
APN (µg/mL)	0.03	0.00	0.044	0.21 *	0.01	0.01	0.01	0.00	0.07	0.00	0.02	0.14 *	0.14 *	0.01
APLN (ng/mL)	0.20 *	0.04	0.10	0.13	0.21 *	0.16 *	0.27 **	0.14 ***	0.13 *	0.01	0.29 ***	0.07	0.48 ***	0.37 ***
RETN (ng/mL)	0.07	0.00	0.10	0.11	0.02	0.00	0.03	0.04 **	0.06	0.02	0.00	0.01	0.00	0.03 **

NW: normal weight; OW: overweight; BMI: body mass index; SBP: systolic blood pressure; DBP: diastolic blood pressure; FBG: fasting blood glucose; TC: total cholesterol; TG: triglycerides; IL-6: interleukin-6; hsCRP: high-sensitivity C-reactive protein; Hcy: homocysteine; Hp: hepcidin; APN: adiponectin; APLN: Apelin; RETN: resistin; Lep: leptin; values are the mean± standard deviation (SD); values are shown as the coefficient of determination (R^2^); *: *p* < 0.05; **: *p* < 0.01; ***: *p* < 0.001; regression lines were plotted for obtaining the values of R^2^ and the values of significance (*p*) using the Statistical Package for Social Sciences (SPSS), version 24.0 for Windows.

**Table 6 jpm-13-00828-t006:** Correlation of systolic blood pressure with other variables on the basis of the body mass index in normal-weight and overweight male university students.

Variable	Coefficient of Determination for the Correlation of Systolic Blood Pressure with Other Variables													
	BMI of NW Subjects (n: 198)								BMI of OW Subjects (n: 192)					
	18–18.9	19–19.9	20–20.9	21–21.9	22–22.9	23–23.9	24–24.9	18–24.9	25–25.9	26–26.9	27–27.9	28–28.9	29–29.9	25–29.9
DBP (mmHg)	0.86 ***	0.97 ***	0.87 ***	0.88 ***	0.67 ***	0.68 ***	0.70 ***	0.75 ***	0.49 ***	0.56 ***	0.82 ***	0.48 ***	0.90 ***	0.72 ***
FBG (mg/dL)	0.00	0.00	0.00	0.015	0.00	0.04	0.05	0.00	0.02	0.00	0.01	0.00	0.25 ***	0.19 ***
TC (mg/dL)	0.05	0.00	0.00	0.00	0.017	0.06	0.01	0.01	0.00	0.05	0.00	0.00	0.01	0.07 ***
TG (mg/dL)	0.00	0.02	0.01	0.08	0.01	0.01	0.03	0.00	0.01	0.00	0.00	0.06	0.08	0.09 ***
IL-6 (pg/mL)	0.01	0.18	0.00	0.01	0.01	0.00	0.01	0.00	0.00	0.01	0.03	0.00	0.018	0.08 ***
hsCRP (mg/L)	0.23 **	0.02	0.03	0.11	0.14 *	0.01	0.11	0.04 **	0.01	0.01	0.03	0.01	0.2 **	0.13 ***
Hcy (μmol/L)	0.23 **	0.12	0.02	0.04	0.09	0.18 *	0.00	0.02	0.01	0.01	0.01	0.03	0.06	0.06 ***
Hp (ng/mL)	0.09	0.00	0.27 **	0.04	0.22 **	0.00	0.00	0.02 *	0.04	0.05	0.17 **	0.00	0.13 *	0.10 ***
APN (µg/mL)	0.08	0.00	0.21 *	0.15 *	0.01	0.00	0.042	0.016	0.00	0.21 **	0.00	0.12 *	0.15 *	0.02 *
APLN (ng/mL)	0.57 ***	0.40 ***	0.73 ***	0.31 **	0.66 ***	0.79 ***	0.83 ***	0.52 ***	0.31 ***	0.53 ***	0.33 ***	0.27 ***	0.53 ***	0.47 ***
RETN (ng/mL)	0.08	0.01	0.00	0.01	0.00	0.03	0.08	0.02	0.10	0.14 *	0.03	0.02	0.02	0.07 ***

NW: normal weight; OW: overweight; BMI: body mass index; SBP: systolic blood pressure; DBP: diastolic blood pressure; FBG: fasting blood glucose; TC: total cholesterol; TG: triglycerides; IL-6: interleukin-6; hsCRP: high-sensitivity C-reactive protein; Hcy: homocysteine; Hp: hepcidin; APN: adiponectin; APLN: Apelin; RETN: resistin; Lep: leptin; values are the mean± standard deviation (SD); values are shown as the coefficient of determination (R^2^); *: *p* < 0.05; **: *p* < 0.01; ***: *p* < 0.001; regression lines were plotted for obtaining the values of R^2^ and the values of significance (*p*) using the Statistical Package for Social Sciences (SPSS), version 24.0 for Windows.

**Table 7 jpm-13-00828-t007:** Correlation of diastolic blood pressure with other variables on the basis of the body mass index in normal-weight and overweight male university students.

Variables	Coefficient of Determination for the Correlation of Diastolic Blood Pressure with Other Variables													
	BMI of NW Subjects (n: 198)								BMI of OW Subjects (n: 192)					
	18–18.9	19–19.9	20–20.9	21–21.9	22–22.9	23–23.9	24–24.9	18–24.9	25–25.9	26–26.9	27–27.9	28–28.9	29–29.9	25–29.9
FBG (mg/dL)	0.01	0.00	0.02	0.00	0.00	0.01	0.00	0.01	0.00	0.01	0.00	0.02	0.20**	0.11 ***
TC (mg/dL)	0.09	0.00	0.00	0.00	0.00	0.00	0.00	0.02 *	0.02	0.01	0.00	0.01	0.00	0.05 **
TG (mg/dL)	0.00	0.02	0.02	0.07	0.01	0.02	0.00	0.01	0.01	0.05	0.00	0.08	0.03	0.06 ***
IL-6 (pg/mL)	0.00	0.21 *	0.00	0.01	0.13	0.03	0.02	0.01	0.01	0.00	0.00	0.00	0.02	0.07 ***
hsCRP (mg/L)	0.19 *	0.02	0.02	0.08	0.13	0.10	0.03	0.04 **	0.00	0.00	0.07	0.05	0.16 *	0.08 ***
Hcy (μmol/L)	0.22 **	0.16 *	0.01	0.01	0.05	0.14 *	0.00	0.021 *	0.04	0.02	0.01	0.05	0.06	0.03 **
Hp (ng/mL)	0.11	0.00	0.26 **	0.07	0.19 *	0.02	0.05	0.01	0.02	0.07	0.10 *	0.04	0.15 *	0.09 ***
APN (µg/mL)	0.04	0.00	0.23 **	0.13	0.00	0.05	0.02	0.00	0.00	0.33 ***	0.00	0.00	0.16 *	0.02
APLN (ng/mL)	0.51 ***	0.31 **	0.63 ***	0.37 ***	0.35 ***	0.75 ***	0.59 ***	0.41 ***	0.26 ***	0.25 ***	0.28 ***	0.27 ***	0.46 ***	0.31 ***
RETN (ng/mL)	0.06	0.00	0.01	0.01	0.00	0.00	0.05	0.02	0.12 *	0.23 **	0.00	0.04	0.03	0.09 ***

NW: normal weight; OW: overweight; BMI: body mass index; SBP: systolic blood pressure; DBP: diastolic blood pressure; FBG: fasting blood glucose; TC: total cholesterol; TG: triglycerides; IL-6: interleukin-6; hsCRP: high-sensitivity C-reactive protein; Hcy: homocysteine; Hp: hepcidin; APN: adiponectin; APLN: Apelin; RETN: resistin; Lep: leptin; values are the mean± standard deviation (SD); values are shown as the coefficient of determination (R^2^); *: *p* < 0.05; **: *p* < 0.01; ***: *p* < 0.001; regression lines were plotted for obtaining the values of R^2^ and the values of significance (*p*) using the Statistical Package for Social Sciences (SPSS), version 24.0 for Windows.

## Data Availability

Not applicable.

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
