# Peer review of "Blood Pressure Correlates with Serum Leptin and Body Mass Index in Overweight Male Saudi Students"

_jpm, 2023, doi:10.3390/jpm13050828_

Round 1
Reviewer 1 Report
This is a work in which the authors analyze the association of serum leptin levels with anthropometric variables such as BMI, BP ...), of which there are controversial data in other populations.
The measurements that the authors make can be interesting, however, in my opinion, the analysis is quite limited. The quality of the work could increase if the authors analyze all the variables they determined and their relationship with different serum leptin levels, for example, by classifying leptin levels in centiles (tertiles or quartiles) and their relationship with other variables.
The data shown in figures 1 and 2 are repetitive, the same data are already reported in table 1, the figures do not provide additional information.
All the figures from 3 onwards, are similar, it would be worth integrating the related figures, but also show another type of statistical analysis: correlation index (not only the R2 value) with other adipokines (levels of apelin, resistin, adiponectin), what risk implies that leptin levels increase (ORs or relative risk) in relation to blood pressure, and the other variables that the authors determined, but that are no longer shown in a subsequent analysis.
Therefore, the discussion section is also very limited. The authors' data seem to be interesting, with better analysis, the results could have much more significance.
Author Response
Reply to Reviewer 1’s Comments:
- We analyzed all variables and determined the relationship among the variables.
- Figures 1 and 2 deleted as suggested.
- We analyzed the correlations using coefficient of determination. Please clarify as we could not understand which specific method/ type of statistical analysis has been suggested. We can convert the coefficient of determination to correlation coefficient, if it is necessary.
- Results for the variations and correlations of all variables are given. The data has been interpreted with the details of all variables in discussion. Hopefully, it will show the importance of the present report.

Reviewer 2 Report
1. The research topic is quite relevant. Leptin is an adipose tissue hormone that has been shown to be associated with several metabolic, inflammatory and other factors known to be involved in the development of hypertension and cardiovascular disease. Several studies have investigated the putative relationship between leptin and hypertension, and their results have been mixed: most studies have not found a direct relationship between leptin and blood pressure, although some associations have been found.
2. The introduction is well substantiated, a comprehensive analysis of the available studies on the relationship between leptin levels and blood pressure is made. At the same time, the discussion is inferior to the introduction in quality. In condemnation, several similar proposals are rather sparingly, that the correlations obtained coincided with certain previous studies, without reasoning on this score. I recommend describing the discussions in more detail, for example, removing some of the statements from the overly detailed introduction and moving them to the discussion section. This will not harm the article.
3. Statistical note: You have used the mean and standard deviation in the presentation of quantitative results, which applies in the case of a normal distribution of quantitative traits. It is necessary to perform a feature test for normal distribution and describe its result. Only if the distribution is normal is it correct to use these methods. In the case of an abnormal distribution of features, it is necessary to present them as medians and quartiles, and instead of analysis of variance, t-test, use other methods of nonparametric statistics.
4. Unfortunately, there is very little relevant literature: 87.8% of the literature sources discussed in the article were published more than 5 years ago, 43.9% - more than 10 years ago or more, and only 12% of the sources of the last five years.
5. Question: why did you only select participants between 18 and 20 years old, this is a very narrow range even in a select cohort of students? Why not take a slightly wider age range to make it easier to extrapolate the results to a larger sample. The men were taken. Please, in the discussion, devote at least a little to the discussion of the gender characteristics of the relationship between leptin and blood pressure. I think that the narrow age range needs to be explained, perhaps described in the study's limitations.
6. Unfortunately, this article does not have sufficient novelty for a journal of this level (Q2). Along with leptin correlations, it describes in detail previously known associations of BMI and BP - these are results that are not new.
minor English correction required
Author Response
Reply to Reviewer 2's Comments
- We agree the leptin has been studied associated with metabolic, inflammatory and a variety of other factors. We found a number of studies describing the association of leptin with obesity, but a little number of studies for investigating the role of leptin in normal healthy and a moderate and high overweight people. Hence, we carried out the present study to understand further the involvement of leptin and its association with body mass index, blood pressure, and other characteristic factors.
- The discussion section is now in more detail. Hopefully it will now serve better for interpreting the association among leptin, body mass index and blood pressure and other variables.
- We apologize that the complete results were not included in the prior submission. We originally categorized the whole data primarily on the basis of body mass index. The detailed results are presented at various levels of body mass index following normal distribution pattern for normal weight and overweight subjects. However, we can make changes for any further suggestion of the respected referee.
- Please, we are including more relevant and recent literature. However, we could not exclude the old but important references in methods.
- Our pilot studies indicated wide difference in the levels of leptin and other factors while considering the age and gender. Hence, we tried to take a start from narrow age range in only male students, though there were also financial reasons for limiting our study to narrow age range and gender. The project funded for the present research by the Umm Al-Qura University was solely for the students entering for the university education (initial two to three years; around 18-20 years age of the students). However, specifying narrow age range and gender are the study’s limitations. To make up for the deficiency, we have recently started studying in male and female students of wider age range, and we will keep in mind the precious guidance of the learned reviewer for our ongoing and future studies.
- Hopefully, the inclusion of other factors in results section would be welcomed. We are thankful to the reviewer for emphasizing to include and analyze the association of other various variables in normal weight and overweight male students. We found that the levels of various factors vary significantly at lower and higher levels of body mass index in normal weight and overweight male students. Further studies may clarify the variations and associations of variables even among the normal weight and among the overweight status by describing as high normal weight and high overweight status.

Round 2
Reviewer 1 Report
It would have been easier to follow the revision of the new version of the manuscript if the authors had highlighted the changes they made in the current version. In addition, it would have been desirable for authors to place the reviewer's comments in the response letter followed by their response to each observation; in this version, it was not possible for me to view the comments I made on the first version of the manuscript.
Although only some of my observations were addressed, I understand and agree with the author's arguments. It is evident that the authors have done an excellent job improving the manuscript, and it could be publishable in its current version, addressing some minor comments.
1. At the bottom of Table 1, indicate the statistical test applied to obtain the p-value. Verify the same in all tables.
2. For clarity, in Tables 3 and 4, I suggest marking the separation between the data for NO and OW subjects, for example, by placing a line under "BMI of NW subjects (n:198)" spanning the corresponding ranges.
3. Table 4 shows the Correlation of serum leptin with other variables. However, I don't see where the leptin values are represented in this table; the label "BMI of NW subjects (n:198)" and the BMI ranges continue to appear in the upper row, similar to Table 3; there is an error in table 4?? Or am I misinterpreting the information? Review. Similar observations are in Tables 5 and 6.
4. The authors wrote: "Below given results in the present section are shown in Table 4". Could you put Table 4 after the description of results in section 3.8? Similar observation for Table 5
5. In the conclusions (last paragraph of the discussion section), the authors should focus on the correlation of leptin with the variables analyzed and the biological importance of this association.
Author Response
Reviewer 1 (Round 2):
- Please see, we indicated the statistical test applied to obtain the P-value, in all Table.
- Kindly see a line added under “BMI of NW subjects (n: 198)” spanning the corresponding range values
- You are not misinterpreting the information, please. In fact, we did a mistake. Kindly see the altered sentence. We wish to clarify that neither the serum levels of leptin in horizontal scale in the Table, nor the levels of other variables in vertical scale are given. We just placed single values of R2 for the plot of leptin against any of the other variables all according to BMI levels in Table 3; and similarly for other Tables (4-6) for respective variables. We made correction in the title of variables placed horizontally (leptin, BMI, SBP, DBP), and hope it clarifies.
- Moved Table 4 and 5, as suggested.
- Added contents for Conclusions after discussion.

Reviewer 2 Report
1. We agree the leptin has been studied associated with metabolic, inflammatory and a variety of other factors. We found a number of studies describing the association of leptin with obesity, but a little number of studies for investigating the role of leptin in normal healthy and a moderate and high overweight people. Hence, we carried out the present study to further the involvement of leptin and its association with body mass index, blood pressure, and other characteristic factors. Reviewer: accepted.
2. The discussion section is now in more detail. Hopefully it will now serve better for interpreting the association among leptin, body mass index and blood pressure and other variables. – Reviewer: accepted. The discussion has been substantially revised and improved, it was interesting and informative to read it.
3. We apologize that the complete results were not included in the prior submission. We originally categorized the whole data primarily on the basis of body mass index. The detailed results are presented at various levels of body mass index following normal distribution pattern for normal weight and overweight subjects. However, we can make changes for any further suggestion of the respected referee. - Reviewer: I didn't quite understand this explanation. Dear researchers, you do not need to provide additional data. It is enough to add to the materials and methods the phrase that the distribution of quantitative characteristics corresponded to the normal one.
4. Please, we are including more relevant and recent literature. However, we could not exclude the old but important references in methods. – Reviewer: accepted, the analyzed literature has been significantly expanded, now 64% of sources are not older than 10 years, 26.4% are not older than 5 years.
5. Our pilot studies indicated a wide difference in the levels of leptin and other factors while considering the age and gender. Hence, we tried to take a start from narrow age range in only male students, though there were also financial reasons for limiting our study to narrow age range and gender. The project funded for the present research by the Umm Al-Qura University was solely for the students entering for the university education (initial two to three years; around 18-20 years of age of the students). However, specifying narrow age range and gender are the study’s limitations. To make up for the deficiency, we have recently started studying in male and female students of a wider age range, and we will keep in mind the precious guidance of the learned reviewer for our ongoing and future studies. Reviewer: аccepted. Dear authors, your explanations have clarified a lot, the limitation section is well finalized.
6.Hopefully, the inclusion of other factors in the results section would be welcomed. We are thankful to the reviewer for emphasizing to include and analyze the association of other various variables in normal weight and overweight male students. We found that the levels of various factors vary significantly at lower and higher levels of body mass index in normal weight and overweight male students. Further studies may clarify the variations and associations of variables even among the normal weight and among the overweight status by describing as high normal weight and high overweight status. – Reviewer: The analysis of other markers added novelty to this study. I welcome these changes.
Author Response
Reviewer 2 (Round 2):
- Happy to hear for the acceptance of our response.
- We are grateful for appreciating our attempt for revising and making it informative.
- Please, we mistakenly sent this reply for the second reviewer, in our last reply. It was actually the suggestion of first reviewer to classify the serum leptin levels into sub-groups and analyze their relationship with other variables. In response to that, it was convenient for us to analyze the whole data at each individual BMI range, though this task took much time. Sorry, we did not add any more data. Rather, we statistically analyzed the same data but in detail at the level of individual BMI values (18-18.9, 19-19.9, 20-20.9… etc).
- Many thanks for the respected reviewer for his comments.
- Glad to hear about our clarifications.
- We are highly thankful to the learned reviewer for welcoming the detailed analysis of other markers.
